# Gene Therapy Strategies for the Treatment of Bestrophinopathies

**DOI:** 10.3390/ijms26199421

**Published:** 2025-09-26

**Authors:** Silja B. Haldrup, Michelle E. McClements, Jasmina Cehajic-Kapetanovic, Thomas J. Corydon, Robert E. MacLaren

**Affiliations:** 1Department of Biomedicine, Aarhus University, 8000 Aarhus C, Denmark; 2Nuffield Department of Clinical Neurosciences, University of Oxford, Oxford OX3 9DU, UK; 3Oxford Eye Hospital, Oxford Hospitals NHS Foundation Trust, Oxford OX3 9DU, UK; 4Department of Ophthalmology, Aarhus University Hospital, 8200 Aarhus N, Denmark

**Keywords:** BEST1, gene therapy, gene editing, bestrophinopathies, CRISPR

## Abstract

The *BEST1* gene encodes a transmembrane protein in the retinal pigment epithelium (RPE) in the eye, that functions as a calcium-dependent chloride channel (CaCC). Pathogenic variants in *BEST1* are the underlying cause for bestrophinopathies, a group of inherited retinal disorders that vary in their pattern of inheritance, clinical appearance, and underlying molecular disease mechanisms. Currently, there are no treatments available for any of the bestrophinopathies, and gene therapy represents an attractive strategy due to the accessibility of the eye and slow disease progression. While gene augmentation may be effective for a subset of bestrophinopathies, others require allele-specific silencing or correction of the disease-causing variant to reconstitute expression of the BEST1 protein. This review aims to give an overview of the clinical diversity of bestrophinopathies and proposes the molecular disease mechanism of the pathogenic *BEST1* variant as an important parameter for the choice of treatment strategy. Furthermore, we discuss the potential of different mutation-specific and mutation-independent CRISPR/Cas9-based gene editing strategies as a future treatment approach for bestrophinopathies.

## 1. Introduction

Gene therapy has the potential to improve the therapeutic landscape of inherited diseases, and treatment approaches for eye conditions are at the forefront of this development due to the eye’s appealing characteristics as a target organ. This includes the eye’s relative immune privilege, compartmentalization, and visual and surgical accessibility [1]. This review focuses on pathogenic variants in the *BEST1* gene, which are causative for bestrophinopathies, a clinically and genetically heterogenous group of retinal degenerative disorders. Phenotypical hallmarks of bestrophinopathies include subretinal vitelliform lesions in the central retina, which progress over time and lead to vision loss and blindness in late stages of the diseases [2]. To date, there are no treatment options available for any of the bestrophinopathies, and novel approaches are therefore urgently needed. Gene therapy strategies may be a feasible treatment option, as the slow disease progression of bestrophinopathies provides a large therapeutic window. Encouragingly, gene augmentation has already proven to be successful for the treatment of autosomal recessive inherited retinal diseases and experienced a major breakthrough with the FDA/EMA approval of Voretigene neparvovec [3,4]. Hence, this approach is particularly promising for autosomal recessive bestrophinopathies with loss-of-function mechanism where supplementation with a wildtype *BEST1* gene may restore sufficient protein function. In contrast, gene augmentation may be applicable to only a subset of autosomal dominant inherited bestrophinopathies, while others require silencing or correction of the disease-causing variant to achieve functional rescue. Here, we provide an overview of the clinical and genetic heterogeneity of *BEST1*-related disorders and highlight the importance of the molecular disease mechanism of the distinct pathogenic variant for the choice of a therapeutic strategy. Furthermore, we discuss the potential of different mutation-specific and mutation-independent CRISPR/Cas9-based gene editing strategies as a treatment approach for bestrophinopathies.

## 2. The *BEST1* Gene and the Bestrophin-1 Channel

The *BEST1* gene is positioned on chromosome 11q13 and consists of 11 exons, of which 10 encode bestrophin-1 (BEST1), a transmembrane protein comprising 585 amino acids (Figure 1A) [2,5]. In the human retina, BEST1 is predominantly localized to the basolateral membrane of the retinal pigment epithelium (RPE). Here, it functions as a calcium-dependent chloride channel (CaCC) that controls the chloride-ion current and maintains the transepithelial potential of the RPE (Figure 1B) [6,7]. Interestingly, research on the high-resolution crystal structure of the eukaryotic BEST1 channel revealed a pentameric configuration with five subunits surrounding a central ion pore [6,8]. Previous studies indicate that the ion pore is “flower vase-shaped” and consists of (i) a funnel-shaped outer entryway, (ii) a slender neck region, (iii) a large inner cavity, and (iv) a narrow aperture [6]. Multiple anion-binding sites within the ion pore facilitate the chloride current of the channel protein and both the neck and aperture region are involved in the gating mechanism that prevents channel leakage [9]. Importantly, the Ca^2+^-clasps near the neck region of each subunit regulate the calcium-dependent opening and closing of the gating apparatus through conformational alterations of the channel protein, thus being crucial for proper channel function. Indeed, especially pathogenic variants associated with dominant inherited bestrophinopathies are known to be clustered around the Ca^2+^-clasp and gating apparatus of the ion pore, while variants leading to recessive inherited bestrophinopathies have been identified outside of this region [6,10,11]. In addition to its function as an CaCC, BEST1 may be involved in (i) intracellular calcium signaling [12], (ii) the regulation of intracellular calcium by influencing the release of Ca^2+^ from endoplasmic reticulum stores [13], and (iii) the regulation of cell volume and homeostasis in RPE cells [14].

## 3. The Clinical Spectrum of Bestrophinopathies

To date, more than 300 distinct variants have been associated with bestrophinopathies [10]. In the Leiden open-source variation database (LOVD) a total of 485 different variants can be found (accessed on 12 January 2024), of which 410 are labeled pathogenic/likely pathogenic or conflicting (Pathogenic/Likely pathogenic vs. Benign/Likely benign vs. uncertain significance) (Figure 2A,B).

The initial diagnosis of bestrophinopathies is based on the phenotypic appearance and a significantly reduced Arden ratio in the electrooculogram (EOG), a sign of comprised RPE function that is considered pathognomonic for these conditions [2].

Notably, bestrophinopathies present in at least five different distinct retinal phenotypes. This includes Best vitelliform macular dystrophy (BVMD), Adult vitelliform macular dystrophy (AVMD) [15,16], autosomal dominant vitreoretinochoroidopathy (ADVIRC), autosomal recessive bestrophinopathy (ARB) [17], and possibly retinitis pigmentosa (RP) [18], which are further described below.

### 3.1. Best Vitelliform Macular Dystrophy

Best vitelliform macular dystrophy (BVMD), or Best disease, is the most common bestrophinopathy with an estimated prevalence between 1:5000 and 1:67,000 [19,20]. The disease was first described by Dr. Friedrich Best in 1905 and follows a dominant inheritance pattern with variable expression [21]. Clinically, BVMD is characterized by the formation of vitelliform lesions within the macula, that slowly progress over time [21,22]. Previous studies have outlined five different stages of disease progression, though they do not occur in all patients (Figure 3A–D).

In the first stage (pre-vitelliform stage), the fundus appears normal, aside from slight changes in the RPE, and the vision is often unaffected [23]. The second stage (vitelliform stage) is characterized by the formation of well-defined yellow, “egg yolk” vitelliform lesions in the macula. At this stage, visual acuity can be slightly decreased, and symptoms including photophobia, metamorphosis, and night blindness may occur [21,24]. In stage three (pseudohypopyon stage), layering of lipofuscin creates the appearance of a pseudohypopyon. However, visual acuity is often not dramatically affected. Further progression to the fourth stage (vitelliruptive stage) occurs when partial resorption and breakup of the vitelliform egg yolk lesion cause a “scrambled egg” appearance of the fundus. This stage is typically accompanied by the substantial loss of vision. The last stage of the disease (atrophic stage) is marked by the death of RPE and photoreceptor loss that causes severe, irreversible vision loss [23]. Notably, in most patients, vision loss progresses slowly, with 75% of the patients younger than 40 years maintaining a visual acuity of 6/12 or better in at least one eye [25]. However, malfunction of the RPE can trigger the growth of choroidal neovascularization (CNV), and subsequent rupture and bleeding of CNV may lead to a rapid decline of vision [26]. Typically, BVMD affects both eyes and presents with unifocal lesions. However, unilateral [26,27], multifocal [28], and asymmetric disease presentation have previously been described. Additionally, some studies have reported a link to other ophthalmological features including hyperopia, potentially resulting from reduced axial length linked to the role of *BEST1* in ocular growth, as well as astigmatism and abnormalities of the anterior segment [22,29,30].

### 3.2. Adult Vitelliform Macular Dystrophy

Adult vitelliform macular dystrophy (AVMD) is an autosomal dominant inherited macula dystrophy with a similar phenotypic appearance to BVMD. However, AVMD is associated with a later age of onset (30–50 years), a slower disease progression, and a generally milder loss of vision. Importantly, due to the later age of onset, it is frequently misdiagnosed as age-related macular degeneration (AMD) [31].

### 3.3. Autosomal Recessive Bestrophinopathy

In contrast to the latter two conditions, autosomal recessive bestrophinopathy (ARB) follows an autosomal recessive inheritance pattern. Hence, ARB patients harbor pathogenic bi-allelic homozygous or compound heterozygous variants in the *BEST1* gene and their parents often present with a normal fundus, EOG, and vision [17,21]. ARB was first described by Burgess et al. and proposed as the null phenotype of *BEST1* in humans [17]. The authors identified two patients with a bi-allelic homozygous nonsense variant (Arg200X) as well as five patients with compound heterozygous mutations with following combinations: p.R141H and p.V317M (one patient), p.L41P and p.P152A (one patient), p.R141H and p.L41P (one patient), and p.D312N and p.M325T (two patients) [17]. More recently, a retrospective case series investigating 18 ARB patients from nine families identified bi-allelic homozygous *BEST1* variants in 16 of the patients, while 2 patients harbored compound heterozygote variants [32]. Clinically, the fundus commonly displays multiple small vitelliform lesions, irregular RPE alterations, and whitish subretinal deposits, mainly found in the macula and midperiphery [21]. Further findings include, drusen and RPE atrophy in the retinal periphery, as well as subretinal fluid accumulation and macular edema [11,33].

### 3.4. Autosomal Dominant Vitreoretinochoroidopathy

Autosomal dominant vitreoretinochoroidopathy (ADVIRC) is a very rare condition (prevalence 1:1,000,000) that follows an autosomal dominant pattern of inheritance with intra-familial phenotypic variability [34,35]. The disease typically manifests during childhood and presents with a circumferential hyperpigmented band between the vortex veins and the ora serrata, a hallmark of the disease. Further findings include, preretinal punctate white opacities, retinal arteriolar narrowing, cystoid macular edema, and fibrillar condensation of the vitreous [36,37]. ADVIRC has also been associated with developmental eye conditions, such as nanophthalmos, microcornea, hyperopia, presenile cataract, a shallow anterior chamber, and optic nerve dysplasia [21,35,37]. Interestingly, previous studies have shown that ADVIRC is associated with *BEST1* variants that cause exon skipping and ultimately lead to shortened and internally deleted isoforms [38].

### 3.5. Retinitis Pigmentosa

Mutations in the *BEST1* gene that are associated with retinitis pigmentosa were initially described by Davidson et al. in 2009 [18]. The authors reported four missense variants in five unrelated families, of which three were autosomal dominant and one was autosomal recessive inherited [18]. The fundus of these patients exhibited dense pigmentary changes and bone spicules in the peripheral retina, retinal gliosis, and vascular attenuation [18]. However, it was recently proposed that these cases of *BEST1*-related retinitis pigmentosa may be misdiagnosed ADVIRC cases [32,39,40]. Alternatively, Dalvin et al. hypothesized that *BEST1*-associated RP may be a multigenic condition, thus requiring additional mutations in other genes [41].

However, despite each of the five *BEST1*-related conditions displaying a characteristic phenotype, bestrophinopathies present with a significant inter- and intra-familiar phenotypic variability, variable expression, and different age of disease onset. Hence, these factors challenge the diagnosis of bestrophinopathies and highlight the importance of genetic testing and thereby identification of the disease-causing variant.

## 4. Molecular Disease Mechanism of Pathogenic *BEST1* Variants

The effect of a specific disease-causing variant on protein level, also known as their molecular disease mechanism, has recently been reviewed thoroughly by Backwell and Marsh [42], and are concisely summarized in the section below.

Briefly, molecular disease mechanisms of pathogenic variants can be divided into loss-of-function, dominant negative, and gain-of-function (Figure 4) [42].

Loss-of-function variants lead to (i) abolished protein synthesis, (ii) a non-functional protein (amorphic), or (iii) a protein with reduced function (hypomorphic). In most cases, loss-of-function variants are recessively inherited, as for instance *BEST1* variants associated with ARB [10]. However, loss-of-function variants may be dominant in cases where haploinsufficiency is at play, hence where both alleles are required to produce a sufficient amount of functional protein. For dominant negative variants, the mutant protein impairs normal protein function by (i) competing with the wildtype protein or (ii) creating a hybrid wildtype–mutant complex, which inhibits the function of the wildtype protein. Notably, for pathogenic *BEST1* variants, assembly of a mutated subunit into the pentameric channel complex has been shown to result in a non-functional or unstable protein, which is subjected to lysosomal degradation or decreases BEST1 channel function [10]. Furthermore, allelic expression imbalance, which is the higher transcription level of mutant allele compared to wildtype allele, has previously been described at the *BEST1* locus of human RPE cells and has shown to promote the dominant negative effect of pathogenic *BEST1* variants [43,44]. Pathogenic variants with a gain-of-function effect are mostly dominantly inherited and alter the encoded protein by either increasing its activity (hypermorphic) or introducing a novel function (neomorphic). Previous studies have provided evidence for a gain-of-function mechanism in ADVIRC-associated variants. For instance, Nachtigal et al. showed increased BEST1-related anion transport for induced pluripotent stem cell (iPSC)-RPE derived from ADVIRC patients (harboring an ADVIRC-associated variant) compared with control cell lines [10].

## 5. Gene Augmentation for the Treatment of Bestrophinopathies

To date, there are no treatment options available for any of the bestrophinopathies, and novel approaches are therefore urgently needed. Notably, the eye offers unique advantages for gene therapy, such as relative immune privilege, compartmentalization, and visual and surgical accessibility. Additionally, several characteristics appoint bestrophinopathies a suitable target for gene therapy: Firstly, given the slow progression of the disease and preservation of central photoreceptor function for a long period, a considerable therapeutic time window exists. Importantly, this treatment window may vary between patients, necessitating careful clinical evaluation of individual disease progression as well as a thorough evaluation of risks and benefits, particularly for patients with retained vision [45]. Secondly, quantifiable therapeutic endpoints like subretinal fluid and vitelliform lesions can readily be assessed with established methods such as optical coherence tomography (OCT) and autofluorescence.

Gene augmentation has already proven to be successful for the treatment of autosomal recessive inherited retinal diseases (IRD) and has experienced a major breakthrough with the FDA approval of Voretigene neparvovec for the treatment of bi-allelic *RPE65* variants [3,4]. While gene augmentation for bestrophinopathies has not yet been evaluated in clinical trials, encouraging preclinical studies in cell and animal models suggest that this strategy may be feasible for recessive inherited bestrophinopathies. For instance, Li and colleagues demonstrated that baculovirus-mediated gene supplementation of wildtype *BEST1* rescued the impaired Ca^2+^-dependent Cl^−^ current in iPSC-RPE cells derived from ARB patients [46] (Table 1). Furthermore, Guziewicz et al. showed that adeno-associated virus (AAV) 2-mediated *BEST1* gene augmentation successfully reversed subretinal lesions and microdetachments and corrected structural alterations within the RPE/photoreceptor interface in a canine BEST1 disease model (Table 1) [47]. Recently, our group developed a quantitative chloride channel conductance assay that enables evaluation of AAV-delivered *BEST1* gene augmentation, thereby offering a novel potency assay for future retinal gene therapy trials [48].

However, since most pathogenic *BEST1* variants follow an autosomal dominant inheritance pattern, it is crucial to determine whether gene augmentation alone is sufficient for these conditions or whether simultaneous disruption of the mutant allele is required. Recently, Sinha et al. hypothesized that increasing the wildtype–mutant BEST1 ratio via gene augmentation may alleviate the disease phenotype in autosomal dominant inherited bestrophinopathies and tested this hypothesis in three iPSC-RPE models harboring various dominant *BEST1* variants. Interestingly, while gene augmentation increased the BEST1 protein levels in all models, calcium-activated chloride channel activity was only restored in two of the three variants. However, CRISPR-Cas9-mediated silencing of the mutant allele established normal channel activity in all variants. This led to the authors proposal to approach treatment of autosomal dominant bestrophinopathies with a two-step strategy: (i) evaluation of gene augmentation in variant-specific patient iPSC-RPEs, and if not successful, (ii) evaluation of gene editing with CRISPR/Cas9-based suppression of the mutant allele (Table 1) [49].

However, given the labor-intensive nature of this approach, identifying characteristics of recessive and dominant bestrophinopathies that may benefit from either gene augmentation or gene editing would be of great interest.

Notably, several studies have proposed the molecular mechanism of pathogenic *BEST1* variants as essential for the choice of the gene therapeutic treatment strategy. For instance, Ji and colleagues demonstrated that AAV2-delivered *BEST1* gene augmentation restored protein expression levels and rescued the impaired Ca^2+^-dependent Cl^−^ current in iPSC-RPEs from patients with dominantly inherited *BEST1* variants with a loss-of-function disease mechanism [50]. Intriguingly, the rescue efficacy of Ca^2+^-dependent Cl^−^ currents was similar to previously treated recessive variants (Table 1) [50].

Furthermore, a recent study by Zhao et al. investigated the therapeutic strategies for *BEST1* loss-of-function and gain-of-function variants in human pluripotent stem cell-derived RPE (hPSC-RPE). Importantly, the results demonstrated that *BEST1* gain-of-function variants in hPSC-RPE could not be rescued with gene augmentation alone (Table 1). However, the authors observed that autosomal *BEST1* variants, previously thought to follow a loss-of-function mechanism, exhibited dominant negative behavior, primarily promoted by allelic expression imbalance [44]. Hence, these variants may be treatable with gene augmentation but potentially require a higher dose to dilute the effect of the mutant protein. Alternatively, including a Woodchuck hepatitis virus posttranscriptional regulatory element (WPRE) in the vector construct could be an option to boost BEST1 expression sufficiently [48].

Thus, dominant and recessive inherited variants with a loss-of-function or dominant negative molecular disease mechanisms may be candidates for gene augmentation, while variants acting through a gain-of-function mechanism require a different treatment strategy (Figure 5).

## 6. CRISPR-Based Treatment Strategies for Pathogenic *BEST1* Variants

The clustered regularly interspaced short palindromic repeats (CRISPR)/Cas9 technology, holds enormous potential for the correction of genetic conditions, and has revolutionized the field of genome engineering and molecular medicine during the last decade. Originally, CRISPR/Cas9 is a natural defense system in bacteria, which provides adaptive immunity against foreign evaders [51]. Since its discovery, researchers have repurposed this system as a versatile tool for gene editing therapy, and in 2020 Jennifer Doudna and Emmanuele Charpentier were awarded the Nobel Prize in Chemistry for its discovery [52]. Encouragingly, CRISPR/Cas 9-based gene editing is showing significant potential to correct genetic disorders directly in the retina, and a phase I/II clinical trial for Edit-101; a CRISPR-based treatment for Leber congenital amaurosis (LCA) type 10 has shown a promising safety profile and improvement in photoreceptor function [53,54].

Essentially, the CRISPR/Cas9 technology consists of two critical elements: (i) the Cas9 nuclease and (ii) a single-guide RNA (sgRNA) that defines the DNA target by a 17–20 nucleotide complementary sequence (Figure 6). Upon reaching the targeted region, the Cas9/sgRNA complex promotes a double-strand break (DSB) in correlation to a protospacer adjacent motif (PAM) site. Subsequently, the cellular DNA repair machinery may attempt to repair the DSB by two major pathways: non-homologous end-joining pathway (NHEJ) or homology-directed repair (HDR) [55]. NHEJ rapidly ligates the DNA ends in a donor template-independent manner, thereby producing unpredictable insertions and deletions (indels). These indels can cause transcriptional frameshifts that induce premature stop codons and/or activate nonsense-mediated decay (NMD), which ultimately lead to degradation of the transcript and silencing of the targeted gene [56]. In contrast, the HDR pathway requires a donor template, thereby enabling the precise correction of a mutation of interest. Recently, the CRISPR toolbox has been further expanded by the discovery of base editing and prime editing, which enable the precise correction of variants in a DSB- and donor template-independent manner (Figure 7).

To address the dominant effect of gain-of-function variants, three different CRISPR-based strategies can be employed: (i) specific silencing of the disease-causing allele, (ii) silencing/knock down of both endogenous alleles and augmentation of the wildtype gene, or (iii) precise correction of the disease-causing variant. These approaches are further elaborated in the section below.

### 6.1. Specific Silencing of Disease-Causing Allele

#### 6.1.1. CRISPR/Cas9-Mediated Gene Silencing

Recently, Sinha et al. utilized the CRISPR/Cas9 technology to specifically silence the disease-causing allele of three different dominant inherited *BEST1* variants. Following lentiviral-mediated delivery of Cas9 and sgRNA targeting the relevant mutant loci, editing rates of up to 80% were detected. Notably, up to 98% of the indels resulted in desired frameshift mutations and thereby targeted silencing. Furthermore, single-cell patch-clamp experiments demonstrated restoration of CaCC activity and thereby functional rescue of BEST1 channel activity (Table 1) [49]. However, despite the potential of this approach for some pathogenic variants, it may not be feasible in cases where the presence of both healthy alleles is crucial and haploinsufficiency is at play. Furthermore, as a mutation-specific strategy, it would require experimental evaluation of a working sgRNA for each variant.

#### 6.1.2. Targeting of Allele-Specific Single Nucleotide Polymorphisms

To achieve mutation independence and allele specificity, a CRISPR/Cas9 strategy targeting single nucleotide polymorphisms (SNPs) specific for the disease allele could be applied. Recently, Salman et al. used this strategy to target AMD-related SNPs in differentiated ARPE-19 cells and demonstrated allele-specific targeting with efficacies of up to 50% [57]. To the best of our knowledge, this strategy has not yet been explored for *BEST1* but may represent an appealing option.

### 6.2. Ablation-And-Replace Strategy

To circumvent the issue of haploinsufficiency, the “ablate and replace strategy” can be employed. This strategy involves (i) non-selective silencing of both endogenous alleles and (ii) simultaneous supplementation of the exogenous wildtype gene. Encouragingly, this strategy has shown promising results in preclinical animal studies as a treatment strategy for autosomal dominant inherited IRDs such as autosomal dominant retinitis pigmentosa [58].

Recently, Zhao et al. utilized a similar strategy that combined transcriptional repression of endogenous *BEST1* with augmentation of wildtype *BEST1* for the treatment of *BEST1* gain-of-function variants. For this purpose, a nuclease-dead Cas9 with a bipartite KRAB-MeCP2 repressor domain (dCas9-KRAB-MeCP2) and a sgRNA targeting both endogenous alleles were delivered to human pluripotent stem cells (hPSC)-derived RPE (hPSC-RPE) cells using two separate baculovirus vectors. The authors demonstrated sufficient repression of the endogenous *BEST1* gene and restoration of the Ca^2+^-dependent Cl^−^ currents by the exogenous wildtype *BEST1* gene (Table 1) [44]. Similarly, our research group has developed a novel mirtron-based “knockdown and replace strategy” for the treatment of rhodopsin-related dominant retinitis pigmentosa (ADRP). Mirtrons are atypical RNA interference (RNAi) effectors that are spliced from transcripts of short introns, thereby bypassing the Drosha/DGCR8 processing step of canonical micro RNAs (miRNAs). Importantly, compared to canonical RNAi effectors, mirtrons have several advantages: (i) Drosha/DGCR8-independent procession results in greater accuracy and may lower the risk of cellular toxicity related to oversaturation of the endogenous microRNA pathway and (ii) expression by Pol-II promotors enables cell-specific transgene expression. In this study, a single vector encoding both rhodopsin-targeting mirtrons and a mirtron-resistant coon-modified version of the rhodopsin coding sequence were delivered via subretinal injection and demonstrated slowing of retinal degeneration in a Rho^23^ knock-in mouse model of ADRP [59]. To the best of our knowledge, this strategy has not been investigated for *BEST1* but could be an attractive option. Further benefits of the “ablate/knockdown and replace strategy” include its potential as a universal treatment strategy for all bestrophinopathies, as it is a mutation-independent approach.

### 6.3. Precise Correction of Disease-Causing Variant

#### 6.3.1. Homology-Directed Repair

The homology-directed repair (HDR)-based CRISPR/Cas9 strategy enables the precise correction of a disease-causing variant in the presence of a DNA donor template. Notably, recent studies have demonstrated precise correction of pathogenic variants in *USH2A* [60] and *ABCA4* [61] in human iPSCs, utilizing a single-stranded oligodeoxynucleotide (ssODN) donor template. However, for many of the preclinical studies conducted in the retina, the HDR-based approach yielded low efficacy. This is possibly due to the postmitotic nature of the retinal cells and the fact that HDR is exclusively active in the late S and G2 phases of the cell cycle [62,63].

#### 6.3.2. Homology-Independent Targeted Integration

An attractive alternative is provided by the homology-independent targeted integration (HITI) strategy, which is based on the NHEJ-repair pathway and enables effective knock-in of a DNA template in postmitotic cells [64]. In contrast to HDR, HITI employs a donor plasmid without homology arms, instead featuring Cas9 cleavage sites that flank the DNA template. This allows for the simultaneous cleavage of both the target DNA sequence and the donor sequence, followed by the integration of donor DNA into the targeted loci (Figure 6) [65].

In the eye, this strategy has been shown to successfully improve vision in a rat model for retinitis pigmentosa [64]. Recently, Meng et al. used HITI to facilitate the precise knock-in of exon 7–11 of the *CYP4V2* gene, a mutation hotspot for Bietti crystalline corneoretinal dystrophy (BCD) [66]. The HITI donor was designed to integrate in intron 6 and contained part of intro 6, the correct sequence of exon 7–11 and a stop codon to mitigate transcription of the disease-causing variants. Encouragingly, the authors showed precise DNA knock-in in BCD-derived patient iPSCs and restauration of viability in iPSC-RPE. Furthermore, HITI-based editing was applied in a humanized *Cyp4v3* mouse model, using dual delivery by subretinal injection. Notably, this led to improved morphology, number and metabolism of RPE and photoreceptors, and improvement of visual function. To our knowledge, the HITI strategy has not been evaluated for the correction of pathogenic *BEST1* variants but could be applied to treat mutation hotspots clustering to specific exons.

However, the Cas9-based gene editing strategies described above rely on the formation of a DSB, which can cause several genotoxic events, including off-target cleavage, chromosomal rearrangement, unintended deletions, chromothripsis, and activation of the p53 pathway [67,68,69,70]. These risks may be overcome by the novel CRISPR-based gene editing tools: base editing and prime editing (Figure 7).

#### 6.3.3. Base Editing

Besides a target-site-defining sgRNA, the base editing technology consists of two components that differ from the above described CRISPR/Cas9-based methods: (i) the Cas9 (D10A) nickase (nCas9), which harbors the RuvC inactivating D10A mutation that promotes nicking of the complementary strand and (ii) a deaminase enzyme inducing the desired base substitution (Figure 7A) [71,72]. Jointly, cytosine base editors (CBEs) and adenosis base editors (ABEs) enable the correction of all four transition mutations and 2 of the 12 transversion mutations [73]. Encouragingly, two recent studies from our group nominated base editing a relevant strategy to correct pathogenic variants in large genes associated with IRDs [74] and *RHO* variants causative for autosomal dominant retinitis pigmentosa [75]. Furthermore, Suh et al. employed base editing to correct a nonsense mutation in *Rpe65* in a murine model in vivo. Subretinal injection of a lentiviral vector encoding base editing components resulted in editing efficacies of 29%, restored RPE65 protein expression, and improved visual function [76]. Recently, Muller and colleagues utilized base editing to correct the most common *ABCA4* mutation (c.5882G > A, p.G1961E), causative for Stargardt disease. Firstly, the authors optimized this strategy in relevant models in vitro, including retinal organoids, induced pluripotent stem cell-derived RPE cells, human retinal explants, and RPE/choroid explants using a dual AAV delivery strategy. Subsequently, they demonstrated high levels of gene correction in mice and non-human primates in vivo, reaching an average editing efficacy of 75% in cones and 87% in RPE cells [77]. To our knowledge, the base editing strategy has not been evaluated for the correction of pathogenic *BEST1* variants. Encouragingly, base editing could theoretically be used to correct a large number of pathogenic *BEST1* point mutations, as for instance the three most frequently reported pathogenic variants from the LOVD (p.R218C, p.R142H and p.A195V). However, the availability of PAM sites within 15 +/− 2 nt from the targeted nucleotide, the target window, and potential bystander edits will need to be taken into consideration and provide a limitation of this technology [78].

#### 6.3.4. Prime Editing

The prime editing technology enables correction of all point mutations, smaller insertions and deletions, and mitigates the risk for bystander edits (Figure 7B) [79]. Intriguingly, prime editing has already been employed to target IRD-causing variants in several preclinical studies. For instance, Jang et al. demonstrated editing efficacies of 6.4% and functional rescue of the disease phenotype in the *rd12* mouse model, after lentiviral-mediated delivery of the prime editing components [80]. Furthermore, Qin and colleagues developed PE^SpRY^, a novel prime editor with unconstrained PAM requirements [81]. Encouragingly, the authors achieved preservation of photoreceptors and rescue of visual function in a mouse model of retinitis pigmentosa, after delivery via a dual AAV system. With its large target scope, prime editing has the potential to correct the majority of pathogenic *BEST1* variants including deletions (i.e., p.I295del). However, due to its large size (>6.3 kB), delivery of the prime editor to the retina provides a challenge, and further evaluation of efficacy and safety are required before moving forward to clinical application [70].

**Figure 7 ijms-26-09421-f007:**
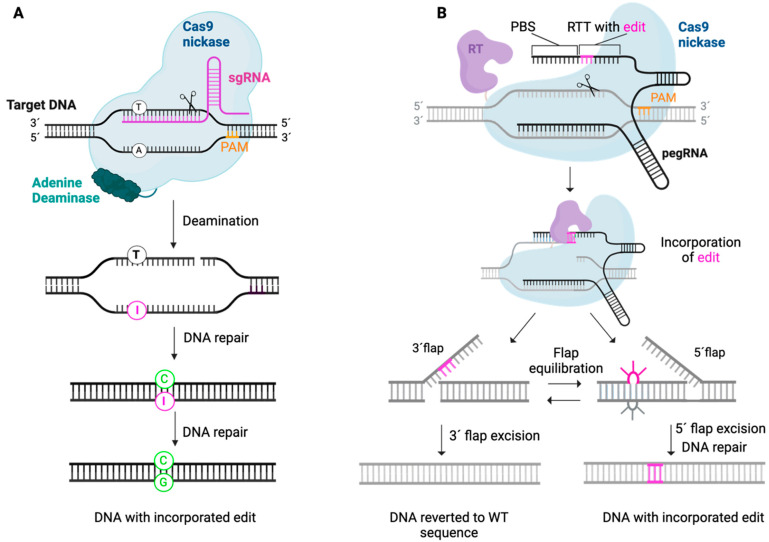
Schematic overview of the base editing and prime editing technology. (**A**) Base editors consist of a sgRNA, a Cas9 nickase, and a deaminase. Upon cleavage of the complementary strand by the Cas9 nickase, the adenine deaminase mediates the transition from A to I (indicated in pink), which is recognized by the cell as G. DNA repair leads to installation of C on the opposite strand (indicated in green) and subsequently installation of G on the deaminated strand (indicated in green). (**B**) Prime editors consist of a Cas9 nickase fused to a reverse transcriptase (RT) and a pegRNA that encodes the desired edit (indicated in pink). Upon cleavage of the non-complementary strand of the target DNA, the PBS of the pegRNA binds to the target DNA and enables the RT to start reverse transcription and thereby incorporation of the edit in this strand. Subsequently, cellular repair mechanisms enable incorporation of the edited DNA 3’flap or the unedited 5’flap, thereby installing the desired edit or reversion to the WT sequence, respectively. A, adenine; C, cytidine; G, guanine; I, inosine; RT, reverse transcriptase; RTT, RT-template; PBS, primer-binding site.

## 7. Conclusions

Bestrophinopathies form a group of inherited retinal diseases with a high complexity regarding their genetic inheritance and clinical appearance. Gene therapy could be a potential treatment strategy for these conditions, and the molecular disease mechanism of the distinct variant could help to further inform the choice of treatment strategy, as illustrated in the proposed workflow from clinical diagnosis to treatment selection in Figure 5. Investigating *BEST1*-related pathogenesis and its correlation to disease phenotypes may further help to choose a relevant treatment. For instance, Nachtigal et al. characterized the cellular and molecular mechanisms underlying the distinct pathologies of ARB, BVMD, and ADVIRC in patient-derived hiPSC-RPE, and developed a classification system for bestrophinopathies, based on these results [10]. Encouragingly, gene augmentation may be a feasible treatment option for *BEST1* variants with loss-of-function disease mechanisms, which constitute the majority of all identified variants [44]. Additionally, preclinical evidence suggests that *BEST1* variants, in which the dominant negative effect is promoted by allelic expression imbalance, may be treatable with gene augmentation, potentially at a higher dose. Lastly, for variants with a gain-of-function mechanism, CRISPR/Cas9 gene editing strategies may provide an interesting alternative. Mutation-independent CRISPR-based strategies are applicable to a wide range of *BEST1* mutations, thereby providing universal treatment for several bestrophinopathies and making them potentially attractive from a commercial perspective. In contrast, mutation-specific strategies require customization for the pathogenic variant of interest, but they offer the potential to restore native gene function.

Collectively, the spectrum of these novel gene therapeutic strategies may provide a useful toolbox to treat a broad range of bestrophinopathies, conditions which are currently untreatable. However, further studies are needed to evaluate their efficacy and safety and optimize retinal delivery. This includes investigation of efficacy in model systems for bestrophinopathies, as for instance patient-derived iPSC-RPE, which enable disease-in-a dish modeling for a specific variant [22]. Furthermore, a thorough off-target analysis should be performed, to ensure high specificity of the gene therapeutic approach. To enable efficient and safe delivery of the therapeutic cargo to the RPE, delivery vehicles should be optimized and tested for the gene therapeutic strategy of choice. AAV-mediated gene augmentation has demonstrated high transduction efficacy and sustained transgene expression in the retina in several clinical trials [82,83] and may be suitable for *BEST1* gene augmentation. In contrast, the large size of gene editing tools including prime editors (>6.3 kb) and base editors (approx. 5 kb) exceed the packaging capacity of a single AAV vector. Alternative strategies to circumvent this limitation include delivery via (i) dual AAV vectors [77,84], (ii) lentiviral vectors [85], (iii) ribonucleoproteins via engineered DNA-free virus-like particles [86,87], or (iv) non-viral delivery vehicles i.e., lipid nanoparticle formations [88]. These approaches have been extensively reviewed by others and are beyond the scope of this work [89,90].

In conclusion, the expanding repertoire of gene therapeutic approaches holds a significant potential to address the treatment of bestrophinopathies, and the molecular disease mechanism of the pathogenic variant may serve as an important parameter for the choice of treatment strategy. However, further research is necessary to enable safe and efficient clinical translation of these novel methods.

## Figures and Tables

**Figure 1 ijms-26-09421-f001:**
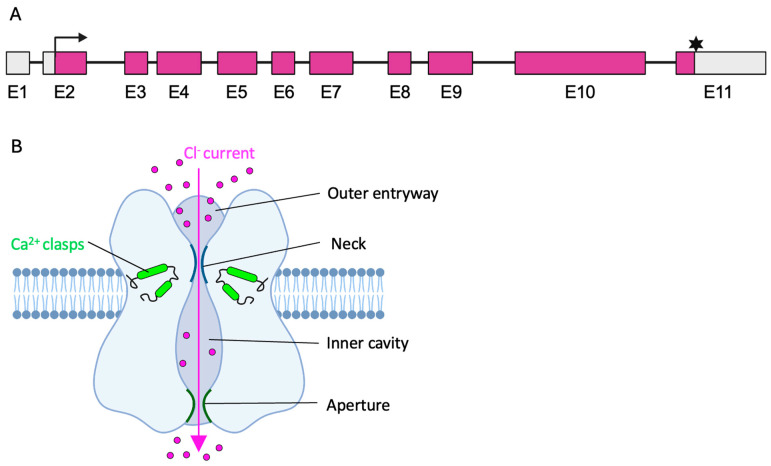
Schematic overview of the *BEST1* gene and BEST1 protein. (**A**) The *BEST1* gene consists of 11 exons (E). Translation of transcription variant 1 (NM_004183.3) starts in exon 2 (Black arrow) and terminates in exon 11 (black asterisk), resulting in a 585 amino acid protein. Translated sequence marked in purple. (**B**) Schematic presentation of the pentameric BEST1 channel. Assembly of the five subunits (blue) forms the ion pore with outer entryway, neck, inner cavity, and aperture. Opening of the gating apparatus is facilitated by the Ca^2+^ clasps and enables the Cl^−^ current through the ion pore.

**Figure 2 ijms-26-09421-f002:**
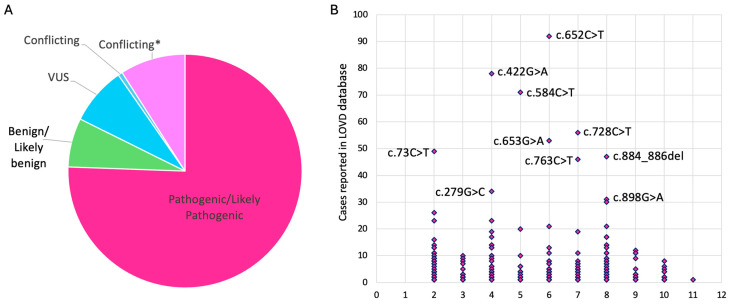
Overview of *BEST1* variants found in the Leiden open-source variation database (LOVD). (**A**) Clinical classification of the identified *BEST1* variants (*n* = 485). There were 366 (75%) variants labeled as Pathogenic/Likely pathogenic; 44 (9%) as conflicting* (Pathogenic/Likely pathogenic vs. benign/likely benign vs. uncertain significance); 3 (1%) as conflicting (Benign/Likely benign vs. VUS); 39 (8%) as variant with uncertain significance (VUS); 33 (7%) as Benign/Likely benign. (**B**) Schematic overview of pathogenic/likely pathogenic/conflicting* *BEST1* variants in respect to their exon position and times reported. The ten most frequently reported variants are annotated with their respective DNA change.

**Figure 3 ijms-26-09421-f003:**
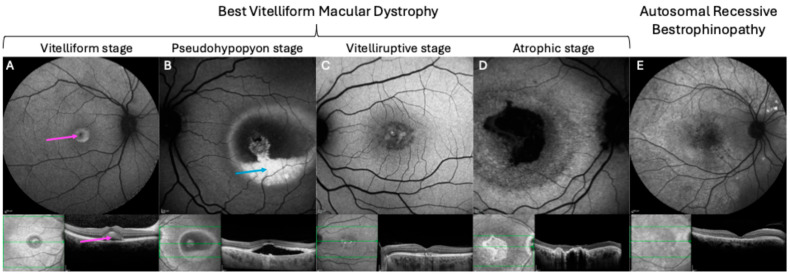
Fundus autofluorescence images (upper row) and optical coherence tomography (OCT) scans (bottom row) depicting the different stages of Best vitelliform macular dystrophy (**A**–**D**) and the typical appearance of autosomal recessive bestrophinopathy (**E**). Green lines on the fundus images in the bottom row indicated the location of the corresponding cross-sectional OCT image shown beside them. Stages of Best vitelliform macular dystrophy include the vitelliform stage (**A**) presenting with well-defined macular lesions (pink arrow), the pseudohypopyon stage (**B**) where layering of lipofuscin creates the appearance of a pseudohypopyon (blue arrow), followed by the vitelliruptive stage (**C**) characterized by a “scrambled egg” appearance, and the atrophic stage (**D**) marked by RPE death and photoreceptor loss. Autosomal recessive bestrophinopathy (**E**) is characterized by multiple small vitelliform lesions and generalized RPE disruption seen as widespread hypoautofluorescent speckling. The images are obtained from Oxford Eye Hospital, Oxford University Hospital NHS Foundation Trust, Oxford.

**Figure 4 ijms-26-09421-f004:**
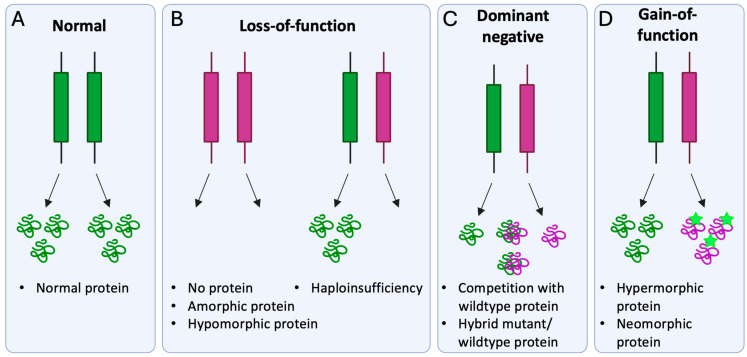
Schematic overview of the molecular disease mechanism and effect at protein level for a (**A**) wildtype gene, (**B**) gene with loss-of-function variant, (**C**) gene with dominant negative variant, and (**D**) gene with gain-of-function variant. Wildtype allele/protein indicated in green; pathogenic allele/protein indicated in purple. Green stars indicate the altered protein.

**Figure 5 ijms-26-09421-f005:**
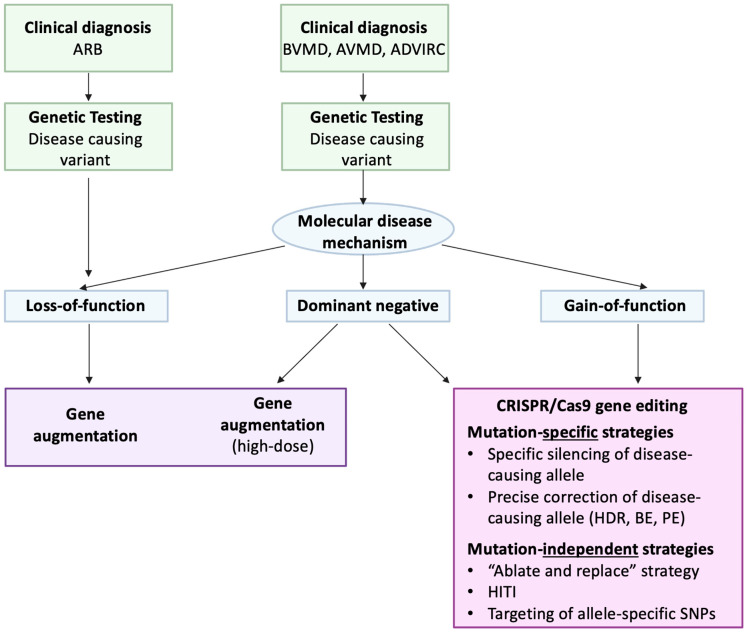
Flowchart from the clinical diagnosis of bestrophinopathies to the choice of treatment strategy. BVMD, Best vitelliform macular dystrophy; AVMD, Adult vitelliform macular dystrophy; ARB, autosomal recessive bestrophinopathy; ADVIRC, autosomal dominant vitreoretinochoroidopathy; HDR, homology directed repair; BE, base editing; PE, prime editing; HITI, homology-independent targeted integration; SNPs, single nucleotide polymorphisms.

**Figure 6 ijms-26-09421-f006:**
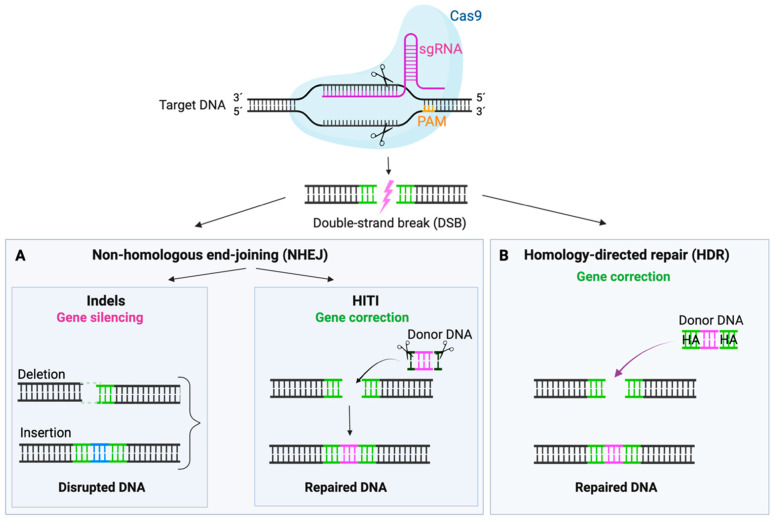
Schematic overview of the CRISPR/Cas9 technology. The CRISPR/Cas9 complex mediates a double-strand break (DSB), which can be repaired via non-homologous end-joining (NHEJ) or homology-directed repair (HDR). (**A**) The NHEJ can facilitate gene silencing via the production of deletions (indicated as dotted lines) and insertions (indicated in blue) (indels) or precise gene correction via utilization of homology-independent targeted integration (HITI). The donor template (indicated in pink) is flanked by Cas9 cleavage sites (indicated with scissors). (**B**) Repair via the HDR pathway facilitates precise gene correction in the presence of a donor template with homology arms (HA).

**Table 1 ijms-26-09421-t001:** Preclinical studies investigating gene therapy for betrophinopathies.

Gene Therapy Strategy	Delivery	Clinical Condition	Inheritance	Disease Model	Outcome	Ref
Gene augmentation	Baculovirus vector	ARB	Autosomal recessive	iPSC-RPE cells	Rescue of Ca^2+^-dependent Cl^−^ current (whole-cell patch clamp recording)	[46]
Gene augmentation	AAV2 vector	ARB (canine model)	Autosomal recessive	Canine BEST1 disease model	Revision of subretinal detachment and microdetachment. Correction of PR/RPE interface	[47]
Gene augmentation	Lentiviral vector	ARB	Autosomal recessive	iPSC-RPE cells	Increased levels of BEST1 protein. Restoration of BEST1 calcium-activated chloride channel activity. Improvement of RPE function (Rhodopsin degradation)	[49]
Gene augmentation	Lentiviral vector	Best disease	Autosomal dominant	iPSC-RPE cells	Increased levels of BEST1 protein. Restoration of BEST1 calcium-activated chloride channel activity. Improvement of RPE function (Rhodopsin degradation)	[49]
CRISPR/Cas9-mediated knock out of mutant allele	Lentiviral vector	Best disease	Autosomal dominant	iPSC-RPE cells	Increased levels of BEST1 protein. Restoration of BEST1 calcium-activated chloride channel activity	[49]
Gene augmentation	AAV2 vector/ Baculovirus vector	BVMD	Autosomal dominant (loss-of-function)	iPSC-RPE	Rescue of Ca^2+^-dependent Cl^−^ current (whole-cell patch clamp recording)	[50]
CRISPR/dCas9-mediated knock down of both alleles + gene augmentation	(i) Baculavirus vector expressing dCas9- KRAB-MeCP2) (ii) Baculovirus vector expressing wildtype *BEST1*		Autosomal dominant (gain-of-function)	hPSC-RPE H1-iCas9 cells *	Rescue of Ca^2+^ dependent Cl^−^ current (whole-cell patch clamp recording)	[44]

* Human pluripotent stem cells (hPSC)-derived RPE (hPSC-RPE) from H1 background hPSC line, carrying an inducible Cas9 cassette. ARB, Autosomal recessive bestrophinopathy; iPSC, induced pluripotent stem cell; RPE, retinal pigment epithelium; AAV, adeno-associated virus; BVMD, Best vitelliform macular dystrophy.

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
