# Peer review of "Gene Therapy Strategies for the Treatment of Bestrophinopathies"

_ijms, 2025, doi:10.3390/ijms26199421_

Round 1

Reviewer 1 Report

Comments and Suggestions for Authors

This is a review describing gene therapy strategies for the treatment of bestrophinopathies. The review is well organized, thorough in its explanation of the problem and the gene therapy strategies that can be used to develop treatments for these patients. The figures are comprehensive and clear. This review will be well received and cited by the inherited retinal degeneration community.

I have a few minor comments aimed to strengthen the review:

-In terms of the genetic mutations, with ~410 pathogenic variants, is it common for BEST1 patients to have more than one mutation that causes the phenotype? Can the authors please clarify and expand on this.

-Can the authors please add these references to the base and prime editing sections appropriately.  

Muller, A., Sullivan, J., Schwarzer, W. et al. High-efficiency base editing in the retina in primates and human tissues. Nat Med 31, 490–501 (2025). https://doi.org/10.1038/s41591-024-03422-8

HoÅ‚ubowicz, R., Du, S.W., Felgner, J. et al. Safer and efficient base editing and prime editing via ribonucleoproteins delivered through optimized lipid-nanoparticle formulations. Nat. Biomed. Eng 9, 57–78 (2025). https://doi.org/10.1038/s41551-024-01296-2

-Can the authors please add one more sentence describing mirtron based knock down and relace strategy.

-In the last sentence the authors say “further studies are needed to evaluate their efficacy, safety, and optimize retinal delivery.” I think it would be beneficial for the authors to touch on what would be an appropriate path to define efficacy and safety for BEST1 gene therapy treatments, especially for gene augmentation and mutation-independent gene editing strategies. With so many mutations, how would you go about proving efficacy across the different genotypes/phenotypes. Is iPSC-RPE enough to show efficacy and then perform safety in a WT large animal model?

-It is recommended to briefly add a section focused on the gene delivery vehicles AAV vs. non-viral especially when comparing gene augmentation vs. gene editing approaches.

Reviewer 2 Report

Comments and Suggestions for Authors

Thank you for allowing me to review the manuscript Gene Therapy Strategies for the Treatment of Bestrophinopathies.

This represents an excellent review of the pathophysiology, genetics and clinical presentation of bestrophinopathies. Would be nice to have photos of all phenotypes.  

The authors discuss theoretical gene therapy modalities based on the molecular disease mechanism of the distinct variants associated with various inheritance patterns.

I suggest adding some information about which stage of the disease the various treatment modalities may be applicable (at least theoretically).

Also, some of the discussed therapeutic modalities have been deployed on other IRDs but not on bestrophinopathies and would be good to highlight that information throughout the manuscript.

Lastly, one challenge with treatment for slow progressive disease is defining reasonable end points to show efficacy and durability of the treatment. I would like the authors to at least speculate on this
